# KIT D816V Mast Cells Derived from Induced Pluripotent Stem Cells Recapitulate Systemic Mastocytosis Transcriptional Profile

**DOI:** 10.3390/ijms24065275

**Published:** 2023-03-09

**Authors:** Marcelo A. S. de Toledo, Xuhuang Fu, Tiago Maié, Eva M. Buhl, Katrin Götz, Susanne Schmitz, Anne Kaiser, Peter Boor, Till Braunschweig, Nicolas Chatain, Ivan G. Costa, Tim H. Brümmendorf, Steffen Koschmieder, Martin Zenke

**Affiliations:** 1Department of Cell Biology, Institute for Biomedical Engineering, RWTH Aachen University Medical School, 52074 Aachen, Germany; marcelo.szymanski@rwth-aachen.de (M.A.S.d.T.);; 2Helmholtz Institute for Biomedical Engineering, RWTH Aachen University, 52074 Aachen, Germany; 3Department of Hematology, Oncology, Hemostaseology and Stem Cell Transplantation, RWTH Aachen University Medical School, 52074 Aachen, Germany; 4Center for Integrated Oncology Aachen Bonn Cologne Düsseldorf (CIO ABCD), 52074 Aachen, Germany; 5Institute for Computational Genomics, RWTH Aachen University Medical School, 52074 Aachen, Germany; 6Institute for Pathology, Electron Microscopy Facility, RWTH Aachen University Medical School, 52074 Aachen, Germany

**Keywords:** mast cell, systemic mastocytosis, *KIT* D816V, induced pluripotent stem cell, disease modeling, RNA-seq, iPS cell differentiation protocol

## Abstract

Mast cells (MCs) represent a population of hematopoietic cells with a key role in innate and adaptive immunity and are well known for their detrimental role in allergic responses. Yet, MCs occur in low abundance, which hampers their detailed molecular analysis. Here, we capitalized on the potential of induced pluripotent stem (iPS) cells to give rise to all cells in the body and established a novel and robust protocol for human iPS cell differentiation toward MCs. Relying on a panel of systemic mastocytosis (SM) patient-specific iPS cell lines carrying the *KIT* D816V mutation, we generated functional MCs that recapitulate SM disease features: increased number of MCs, abnormal maturation kinetics and activated phenotype, CD25 and CD30 surface expression and a transcriptional signature characterized by upregulated expression of innate and inflammatory response genes. Therefore, human iPS cell-derived MCs are a reliable, inexhaustible, and close-to-human tool for disease modeling and pharmacological screening to explore novel MC therapeutics.

## 1. Introduction

Mast cells (MCs) are rare but key cells of the hematopoietic system with a pivotal role in innate and adaptive immune responses [1,2]. They act as sentinels of the immune system and are frequently located in tissues, which are directly exposed to environmental challenges, such as the gastro-intestinal, respiratory, urogenital tracts, and skin [3]. Following activation, MCs modulate immune responses by secreting a vast number of mediators, including cytokines, chemokines, proteases among others, or by direct cell–cell contact [4]. As a result, MCs efficiently respond to infection or injury by recruiting and modulating the activity of effector cells. However, MCs are still a rather poorly studied cell of the hematopoietic system and the origin of MC precursors during embryogenesis and in the adult are currently being explored [5,6]. Recently, independent studies showed that peripheral blood circulating CD34^+^KIT^+^FcER1^+^ cells are MC precursors that can be differentiated in vitro into mature functional MCs [7,8,9].

In line with the pleiotropic functions of MCs, their abnormal proliferation, infiltration, and accumulation in organs, such as the spleen, liver, skin, or bone marrow (BM), drives a heterogenous pathology, referred to as systemic mastocytosis (SM) [10,11]. The burden of MC infiltration in the different affected organ(s) correlates with the amount of secreted MC mediators that cause tissue damage and consecutive development of symptoms, further contributing to disease heterogeneity. Advanced entities of this disease, such as aggressive systemic mastocytosis (ASM) and SM with associated hematological neoplasm (SM-AHN), may evolve to mast cell leukemia (MCL), a disease with strikingly poor overall survival [11].

The key molecular event in the onset and development of SM is the *KIT* D816V mutation, identified in up to 90% of SM patients [11,12]. This mutation renders the KIT tyrosine kinase receptor constitutively active (independent of its natural ligand, stem cell factor, SCF), affecting key cellular pathways in MC biology, such as proliferation, development, and activation [13]. Interestingly, the *KIT* D816V mutation has been described throughout the hematopoietic lineage, from hematopoietic stem cells to lymphoid and non-MC myeloid lineages, which may further contribute to disease heterogeneity [14,15,16,17]. Co-occurring mutations in ASXL1, RUNX1, and SRSF2 have been identified and correlated with disease progression, severity, and overall survival [18]. KIT, as a key cytokine receptor for MCs with strong pathological implications in SM development, has become a valuable therapeutic target [19]. Accordingly, several KIT-targeting tyrosine kinase inhibitors are in clinical use for SM, such as midostaurin and avapritinib. However, to date, the only therapy with curative potential is allogenic stem cell transplantation [20,21].

Most of our knowledge of MC biology and pathology derives from murine models, human and murine ex vivo-differentiated MCs, and human immortalized MC lines [22,23,24,25,26,27,28]. Although valuable, these models fail to properly recapitulate many aspects of human MC development, biology, and the genetic disease heterogeneity found in SM patients. In this context, patient-specific induced pluripotent stem (iPS) cells are an attractive model for studying MC biology in health and disease. iPS cells enable the faithful recapitulation of the individual patient’s genetic background, preserve the pathogenic molecular lesion, and additionally, the patient’s hematopoietic clonal composition [29,30]. By implementing tailored protocols, these iPS cell lines are efficiently differentiated toward the hematopoietic lineage, including MCs [31,32,33,34]. Additionally, iPS cells are a virtually inexhaustible cell source, thus overcoming material limitations when studying rare cell types such as MCs. Further to this, patient-derived iPS cell lines can be precisely engineered with the CRISPR/Cas9 editing technology [34,35,36,37]. Specific pathology-related molecular lesions can be introduced or repaired, further extending the applicability of iPS cells in disease modeling and drug screening studies. 

Here, we built on our previous work on patient-specific iPS cell lines with *KIT* D816V mutation [33] and developed a robust protocol for iPS cell differentiation toward MCs in a feeder-free system with minimal xenobiotic components. The iPS cell-derived MCs obtained are functional and degranulate upon IgE cross-linking. They also recapitulate pathological features of *KIT* D816V MC, such as abnormal surface expression of CD25 and CD30, which is used as clinical criteria for the SM diagnosis [11]. In addition, mutated MCs display a more activated phenotype showing higher expression of FcER1 and MRGPRX2. Finally, global gene expression analysis of *KIT* D816V MCs by RNA-Seq revealed an upregulation of interferon (IFN) signaling, innate and adaptive immune response, and inflammatory response pathways. The observed transcriptional profile faithfully recapitulates that of SM patient-derived MCs. Altogether, we developed a robust protocol for the generation of iPS cell-derived MCs as a reliable tool for the study of MC biology in health and disease.

## 2. Results

### 2.1. MCs Are Efficiently Differentiated from KIT D816V iPS Cells in a Feeder-Free Spin EB Culture

To exploit the full potential of iPS cells as a powerful tool to study MC biology and pathological mechanisms, a robust differentiation protocol is required. In addition, short culture time, low cost, and use of xenobiotic-free components are desirable features to allow the widespread implementation and translation of iPS cell differentiation protocols. With that in mind, we established a three-step protocol for the differentiation of feeder-free iPS cell cultures toward the hematopoietic lineage and further toward MCs (Figure 1A,B). We used iPS cell lines carrying the *KIT* D816V mutation or control iPS cell lines (with unmutated *KIT*) derived from SM patients (listed in Table 1) established in our previous work [32,33]. These SM iPS cell lines enabled us to validate our protocol in a disease-specific background and to investigate how iPS cell-derived MCs are affected by the *KIT* D816V mutation. 

In the first step, embryoid body (EB) formation is initiated by the spin-EB method, where forced aggregation of single-cell iPS cell suspension is achieved by centrifugation in a 96-well format. Mesoderm commitment followed by hematopoietic differentiation is induced by the sequential application of a defined combination of cytokines (Figure 1A and Table 2). Hematopoietic cells start to be released from EB on day 8–10. On day 14, for each 96-well plate, an average of 2.786 ± 2 × 10^6^ hematopoietic cells (mean ± SD, n = 7) were obtained for *KIT* D816V iPS cells, while an average of 3.83 ± 3.3 × 10^6^ hematopoietic cells (mean ± SD, n = 9) were obtained for control iPS cells (Figure 1C, left graph).

To synchronize the kinetics of MC differentiation of *KIT* D816V and control cells, we purified CD34^+^ hematopoietic stem/progenitor cells (HSPC) by immunomagnetic bead selection on day 14. Routinely, 50.67 ± 22.6% of *KIT* D816V hematopoietic cells (mean ± SD, n = 7) were CD34^+^, while for control cells, the CD34^+^ fraction was 40.25 ± 13.7% (mean ± SD, n = 9, Figure 1C, right graph).

The second step promotes the hematopoietic expansion of CD34^+^ HSPC for 10–14 days in a myeloid cell supporting medium (supplemented with IL-3, hyper-IL-6 [38], Flt3L, and SCF, Table 2). At the end of this step, MC progenitors were obtained, quantified, and isolated by FACS sorting for CD45^+^KIT^high^ cells (Figure 2A). A homogenous population of MC progenitors with multi-lobulated nuclei and with low cytoplasmic granularity was observed by acidic toluidine blue staining (Figure 2A, right panel). Recapitulating our previous data [33] and in line with the aberrant expansion of *KIT* D816V MCs in SM patients, we observed a trend toward an increased number of mutated CD45^+^KIT^high^ MC progenitors in comparison to control cells (Figure 2B). *KIT* D816V MC progenitors also displayed enlarged cell size, shown by FSC analysis in comparison to control cells, while no difference in cell granularity was noted (Figure 2C). The enlarged cell size observed for *KIT* D816V MCs may indicate a more advanced maturation status, as MCs have been shown to increase in size upon maturation [39]. Highlighting the potential of iPS cells to generate substantial amounts of rare cell types, such as MCs, starting with a 96-well plate seeded with 5 × 10^5^ iPS cells, we routinely obtained 3.6 ± 2 × 10^6^
*KIT* D816V MC progenitors (n = 8) and 1.9 ± 1.8 × 10^6^ control MC progenitors (n = 10; Figure 2B, right graph).

### 2.2. KIT D816V iPS Cell-Derived MCs Recapitulate Features of SM MCs

In the final step of our protocol, MC maturation was performed by culture for additional 60–70 days in the presence of hyper-IL-6 and SCF (Table 2). After this step, a high purity (≥95% CD45^+^KIT^high^) MC population was obtained. Acidic toluidine blue staining of *KIT* D816V and control MCs revealed cells displaying metachromatic granules in the cytoplasm and dense non-lobulated nuclei. MCs were also tryptase positive (Figure 2D). *KIT* D816V and control MCs showed similar granularity and size as demonstrated by FSC and SSC analysis (Figure 2E). Flow cytometry analysis further revealed higher surface expression of CD25 and CD30 in *KIT* D816V MCs in comparison to control MCs (Figure 2F). These results are in agreement with clinical data where aberrant expression of either of these markers is used as a further criterion in the SM disease diagnosis [10]. CD2 expression was not detected on the surface of *KIT* D816V or control MC. CD33, a pan-myeloid marker also used as a criterion in SM disease diagnosis and a potential therapeutic cell surface target, was highly expressed in both, *KIT* D816V and control MCs (Figure 2F) [11,40].

**Figure 2 ijms-24-05275-f002:**
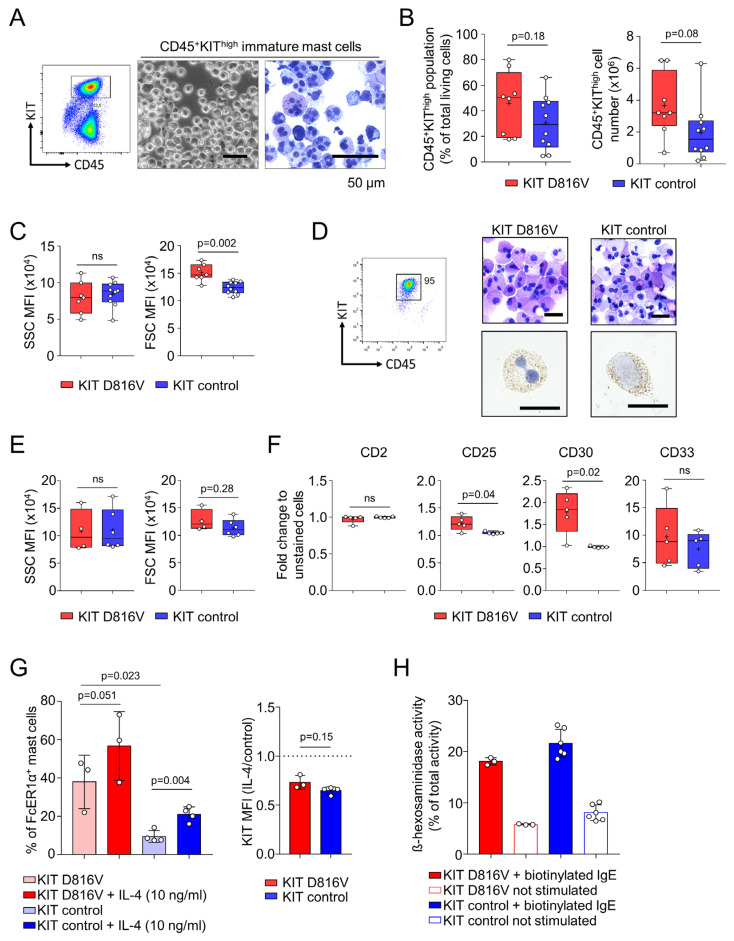
iPS cell-derived *KIT* D816V MCs recapitulate systemic mastocytosis features. (**A**) Exemplary dot plot (**left panel**) of fluorescence-activated cell sorting (FACS) showing the CD45^+^KIThigh immature MC population. After FACS sorting, a homogeneous cell population was observed, as shown by phase contrast microscopy (**middle panel**, scale bar: 50 µm). Acidic toluidine blue staining (**right panel**, scale bar: 50 µm) revealed immature MCs with multi-lobulated nuclei and hypogranular cytoplasm. (**B**) CD45^+^KIT^high^ population size on the day of FACS sorting (**left panel**, Welch’s *t*-test, *p* = 0.18, *KIT* D816V, n = 8; control, n = 10) and a total number of CD45^+^KIT^high^ cells sorted (**right panel**, Welch’s *t*-test, *p* = 0.08, *KIT* D816V, n = 8; control, n = 10). (**C**) *KIT* D816V CD45^+^KIT^high^ cells show no difference in granularity (**left panel**, side scatter, SSC) but increased size (**right panel**, forward scatter, FSC) in comparison to control cells (Welch’s *t*-test, *p* = 0.6211 (**right panel**), *p* = 0.002 (**left panel**), *KIT* D816V, n = 7; control, n = 10). (**D**) Exemplary FACS dot plot of iPS cell-derived MCs after 60–70 days of maturation (**left panel**). Acidic toluidine blue staining of *KIT* D816V and control MCs after maturation stage. Cells with metachromatic granules in the cytoplasm and dense and non-lobulated nuclei are observed (**middle and right upper panels**, scale bar: 20 µm). Mature *KIT* D816V and control MCs are positive for tryptase (**middle and right bottom panels**, scale bar: 20 µm). (**E**) *KIT* D816V and control MCs show similar granularity (**left panel**, side scatter, SSC) and size (**right panel**, forward scatter, FSC) after 60–70 days of maturation. (**F**) iPS cell-derived *KIT* D816V MC recapitulate aberrant expression of CD25 and CD30 (Welch’s *t*-test, CD2 *p* = 0.35, CD25 *p* = 0.04, CD30 *p* = 0.02, CD33 *p* = 0.46, *KIT* D816V n = 5, control n = 5). (**G**) In the **left panel**, IL-4 treatment (10 ng/mL) for 4 days led to increased FcER1α surface expression in *KIT* D816V MCs (fold increase of 1.53 ± 0.25, Welch’s *t*-test, n = 3, *p* = 0.051) and control MCs (fold increase of 2.31 ± 0.34, Welch’s *t*-test, n = 3, *p* = 0.0049). *KIT* D816V MCs show higher FcER1α surface expression (One-way ANOVA—multiple comparison, *KIT* D816V and control n = 3, *p* = 0.023). In the **right panel**, IL-4 treatment (10 ng/mL) for 4 days led to the downregulation of cell surface KIT expression in *KIT* D816V and control MCs (Welch’s *t*-test, *KIT* D816V, and control n = 3, *p* = 0.15). (**H**) *KIT* D816V and control iPS cell-derived MCs degranulate upon FcER1α activation by IgE crosslinking. β-Hexosaminidase activity in the supernatant of biotinylated IgE-stimulated cells is higher than for not stimulated cells (*KIT* D816V n = 1, control n = 2, see Appendix A). For (**B**,**C**,**E**,**F**), box and whiskers plots show minimum and maximum values, and “+” indicates the mean value. ns indicates not significant.

Interleukin 4 (IL-4) is a cytokine produced by lymphocytes, eosinophils, basophils, and MCs. IL-4 acts on MCs, affecting their proliferation and the profile of the mediators they secrete, and upregulates FcER1α surface expression while downregulating the KIT surface expression [41,42,43]. In this context, we investigated the IL-4 effect on FcER1α and KIT surface expression in mature iPS cell-derived MCs. We observed that *KIT* D816V MCs have higher FcER1α surface expression in comparison to control cells, indicating a more mature or activated status. Upon IL-4 treatment, FcER1α surface expression was increased in both *KIT* D816V and control MCs with a more pronounced effect on the latter (Figure 2G, left panel). In agreement with previously reported data, KIT surface expression was downregulated in both, *KIT* D816V and control MC, upon IL-4 treatment (Figure 2G, right panel). We further investigated if iPS cell-derived MCs were able to degranulate upon FcER1α activation by IgE. In line with the observed surface expression of FcER1α, both *KIT* D816V and control MCs degranulated upon IgE stimulation, as shown by β-hexosaminidase activity in the culture supernatant (Figure 2H and Appendix A).

Transmission electron microscopy (TEM) of *KIT* D816V and control MCs revealed MC-specific ultrastructure (Figure 3 and Appendix A). In line with reported ultrastructural data for human MCs, iPS cell-derived MCs displayed thin cytoplasmic projections, modest Golgi-associated endoplasmic reticulum, and cytoplasmic membrane-delimited secretory granules with distinct electron densities and granularity. Of note, “scroll-like” granules, typical of MCs, were also observed [44,45]. In *KIT* D816V MCs, we also observed a pronounced enlargement of secretory granule compartments with reduced granular content (Appendix A). These structures are indicative of MC activation that may lead to secretory granule membrane fusion and piecemeal degranulation, a process in which MCs release their granule contents but retain the granule membranes within the cytoplasm [44].

Altogether, our data show that the protocol for iPS cell differentiation toward MCs described here generates a large amount of functional MCs and very well recapitulates features associated with *KIT* D816V neoplastic MCs in SM.

### 2.3. KIT D816V iPS Cell-Derived MCs Recapitulate the Gene Expression Profile of SM MCs

To gain further information on the functionality and the impact of the *KIT* D816V mutation on iPS cell-derived MCs, we performed global gene expression analysis on mature *KIT* D816V and control MCs by RNA-Seq. Confirming their MC identity, 11 out of 14 MC signature genes described by Motakis and colleagues were highly expressed in iPS cell-derived MCs (Figure 4A and Appendix A) [46].

Both *KIT* D816V and control MCs showed high expression of tryptase genes (*TPSAB1*, *TPSB2,* and *TPSD1*) and enzymes associated with the biosynthesis of MC mediators (*HDC*, *HPGDS*, *LTC4S*). In line with our flow cytometry data (Figure 2G) and previous work, two subunits of the FcER1 (*FCER1A*, *MS4A2*) showed higher expression in *KIT* D816V MCs than control (Figure 4A and Appendix A). In addition, we observed higher expression of Mas-related G protein-coupled receptor X2 (*MRGPRX2*) in *KIT* D816V MCs (Appendix A). MRGPRX2 has a pivotal role in IgE cross-linking-independent MC activation [47]. We further observed high expression of gene coding for granule membrane-associated proteins such as *VAMP8* and *VAMP3,* in *KIT* D816V MCs (Appendix A). Expression of lysosomal associated membrane proteins *LAMP3* (CD63), *LAMP1* (CD107a), and lipid raft resident transmembrane adaptor molecule *LAT2*, all involved in MC degranulation, was also higher in *KIT* D816V cells compared to control (Appendix A). 

Gene set enrichment analysis (GSEA) revealed an upregulation in *KIT* D816V MCs of pathways related to IFN-α, -β, and -γ signaling, innate and adaptive immune responses (including viral response), inflammation, cytokine signaling, and MC activation (Figure 4B,C, Appendix A). This observation is in line with the work of Teodosio and colleagues, who described a similar gene expression profile in MCs isolated from the BM of *KIT* D816V SM patients [48]. In this context, we observed a strong enrichment of genes found deregulated in primary SM MCs in the *KIT* D816V MC transcriptome (Figure 5A). We further validated selected key genes by RT-qPCR and confirmed in iPS cell-derived *KIT* D816V MC trend toward the upregulation of interferon signaling/inflammatory response pathways (*IFIT2*, *STAT1,* and *CCL23*, Figure 5B). Next, we investigated the expression of genes involved in pathways that were also found to be deregulated in primary SM MCs. We observed significant upregulation in *LAT2* (MC degranulation), *APOL1* (lipid metabolism), *CHF* and *SERPING1* (complement regulation), and *CASP1* (apoptosis) in *KIT* D816V iPS cell-derived MCs (Figure 5C). In conclusion, transcriptome analysis confirmed the identity of iPS cell-derived MCs and, additionally, revealed a more activated phenotype of *KIT* D816V MC. Moreover, *KIT* D816V MCs derived from iPS cells recapitulate the gene expression profile of primary *KIT* D816V SM MCs, characterized by the upregulation of pathways involved in innate and adaptive immune responses and inflammation. 

## 3. Discussion

Here, we presented a novel and robust protocol for the differentiation of patient-specific iPS cells toward the MC lineage in a feeder-free culture system. By using previously established and characterized SM patient-derived *KIT* D816V and control iPS cell lines, we built further on our efforts to exploit patient-specific iPS cells as a model for SM [33]. The panel of iPS cell lines used in this work allowed us to confirm the robustness of our protocol, as MCs were obtained from nine different iPS cell lines generated from three patients (Table 1). In addition, the only xenobiotic supplement in our protocol is bovine serum albumin (BSA), used in the first two days and replaced afterward by human serum albumin (HSA). The defined composition of our basal medium, especially at the myeloid expansion and MC maturation stage, allows the precise assessment of factors influencing MC development and function. Another highlight of our protocol is the use of low-cost basal medium (Table 2), in contrast with commercially available media and kits commonly used in iPS cell differentiation protocols (e.g., StemDiff, StemPro34, and StemSpan) [49,50]. With this basal medium and optimized cytokine cocktails, MCs are efficiently produced at the end of three steps: first, we rely on feeder-free iPS cell culture and mesoderm-hematopoietic commitment through the formation of 3D EB structures in the presence of BMP-4, VEGF, and SCF; second, myeloid cell expansion is performed in the second step, driven by SCF, IL-3, Flt3L, and hyper-IL-6; and third, MC maturation is achieved in the presence of SCF and hyper-IL-6. 

In our protocol, CD34^+^ MACS isolation of HSPC allows the synchronization of differentiation kinetics, a key step when evaluating the impact of molecular lesions, culture conditions, or drugs on MC development and function. Similarly, CD45^+^KIT^high^ FACS sorting enables the maturation step to occur in a homogenous MC population, without the potential interference of other hematopoietic cell types and the cytokines they might secrete. However, omitting CD34^+^ MACS selection does not compromise MC differentiation, but it is recommended to deplete for tissue culture plastic-adherent macrophages during the myeloid/MC maturation step.

The *KIT* D816V iPS cell-derived MCs obtained via our protocol recapitulate phenotypic features of SM MCs, such as abnormal expansion and aberrant surface expression of CD25 and CD30. Furthermore, mutant MCs showed higher FcER1 surface expression in comparison to control cells. The gene coding for MRGPRX2, a pivotal receptor in non-IgE-mediated MC activation and responsible for the connection between MC-mediated immune regulation and neurologic stimuli, showed higher expression in *KIT* D816V MCs than in control MCs [47]. TEM analysis of iPS cell-derived MCs revealed MC typical cytoplasmic secretory granules and indicated a more activated degranulating status of *KIT* D816V MCs. The observation of large membrane delimited structures with low granule content, more abundant in *KIT* D816V MCs, suggests the occurrence of piecemeal degranulation, a process of MC degranulation upon activation commonly observed under inflammatory conditions [3]. Altogether, our data indicate a more activated status of *KIT* D816V MC, in agreement with SM clinical features [51].

Teodosio and colleagues reported a common gene expression profile for MCs isolated from the BM of *KIT* D816V patients with different SM entities [48]. Global gene expression analysis of *KIT* D816V and control iPS cell-derived MCs recapitulated this transcriptional profile in mutated cells. GSEA revealed an upregulation of pathways related to innate and adaptive immune response, anti-viral response, cytokine signaling, IFN signaling, inflammatory response, and MC activation in *KIT* D816V MCs. In the BM of SM patients, MCs are in close and constant interaction with BM niche components (e.g., stroma cells, vasculature, and other hematopoietic cells) and soluble mediators that are mostly lacking in our in vitro differentiation system. Therefore, the fact that in vitro generated *KIT* D816V iPS cell-derived MCs recapitulate the key transcriptional features of *KIT* D816V SM patient-derived MCs strongly suggests that this signature is driven by the *KIT* D816V mutation and the constitutive activation of KIT signaling pathways. KIT signaling is not only essential for MC development and survival but has also been shown to impact MC function and modulate cell migration and adhesion [52]. Moreover, in synergy with FcER1 engagement, KIT signaling enhances MC degranulation and production of cytokines, such as IL-6 [53,54]. Hence, the constitutive activation of oncogenic *KIT* D816V likely leads to the deregulated MC maturation and activation features we observed in our *KIT* D816V iPS cell-derived MCs, such as high *FcER1A* and *MRGPRX2* expression and upregulated cytokine signaling and MC activation pathways.

MCs originate from BM but mature in their final tissue of residence, which has a direct impact on the mature MC phenotype and function [3]. Therefore, we envision that better models to study MC biology, disease, and therapeutic targeting need to be developed. An attractive approach consists of 3D coculture systems of iPS cell-derived immature MCs with fibroblasts, endothelial, and mesenchymal cells in 3D matrixes. These niche-mimicking constructs should provide a model closer to the in vivo situation, such as MC niches in the connective tissue or the BM niche hosting SM MCs. In this context, MC maturation may be achieved by niche-secreted factors, such as SCF and IL-6. In addition, pathological scenarios can be recapitulated by using iPS cell-derived MCs harboring disease-relevant mutations (e.g., *KIT* D861V, *JAK2* V617F) or by adding effector molecules, such as pro-inflammatory cytokines. In summary, our protocol reported here paves the way for implementing patient-specific iPS cell-derived MCs as reliable tools to investigate MC biology, pathomechanisms, and drug response, overcoming the cell number and availability limitations faced when using primary MC. 

## 4. Materials and Methods

### 4.1. The iPS Cell Lines and iPS Cell Culture

The iPS cell lines used in this study were generated and characterized in our previous work and are listed in Table 1 [32,33]. Patient 1 and 2 iPS cell lines were generated from peripheral blood mononuclear cells, and patient 3 iPS cell lines were generated from BM mononuclear cells. The iPS cells were cultured with StemMACS iPS-Brew XF (Miltenyi Biotec, Bergisch Gladbach, Germany) on 6-well plates coated with Matrigel (Corning, USA) following the manufacturer’s recommendations. Culture passaging was performed with Accutase (PAN-Biotech GmbH, Aidenbach, Germany) or 0.2 mM EDTA PBS solution (both from Gibco, USA).

**Table 1 ijms-24-05275-t001:** List of iPS cell lines used in the present work.

Patient	Diagnosis	*KIT* Mutational Profile	iPSC Line	Additional Mutations in Each iPSC Line	Reference
1	ASM	D816V	iPSC_*KIT* D816V #1	…	Toledo et al., 2021 [33]
			iPSC_*KIT* D816V #2	*NFE2* (p.Glu261Alafs*3)	
			iPSC_*KIT* D816V #3	*NFE2* (p.Glu261Alafs*3)	
			iPSC_control #1	*NRAS* (p.Gly12Asp), *TET2* (p.Cys973Alafs*34)	
			iPSC_control #2	*NRAS* (p.Gly12Asp), *TET2* (p.Cys973Alafs*34)	
2	MCL	D816V	iPSC_*KIT* D816V #4	…	Toledo et al., 2021 [33]
			iPSC_control #3	…	
3	MCL	S476I	iPSC_control #4	...	Atakhanov et al., 2022 [32]
			iPSC_control #5		

### 4.2. The iPS Cell Differentiation toward MCs

The iPS cells were cultured as described above up to a confluency of 70–80%. On day 0 of the first step (Figure 1A), cells were treated with Accutase (PAN-Biotech GmbH, Aidenbach, Germany) for 4 min at 37 °C, followed by a wash step with KO-DMEM (Gibco, USA). A total of 5 × 10^5^ iPS cells were seeded per well in a 96-well U-bottom suspension culture plate (Greiner, Frickenhausen, Germany) in d0 Spin EB medium (please refer to Table 2 for medium composition) followed by centrifugation at 360× *g* for 7 min. Cells were incubated at 37 °C and 5% CO_2_ throughout the entire differentiation protocol. On day 2, day 2 Spin EB medium was added to the wells. From day 3 onwards, 50 µL of the medium was removed from the wells, and 50 µL of fresh medium was added (Table 2). On day 14, spin EB and hematopoietic stem/progenitor cells (HSPC) derived thereof were harvested, passed through a 40 µm cell strainer (Greiner), and subjected to CD34^+^ magnetic-activated cell sorting (MACS) as described below. In the second step, CD34^+^ HSPC (≤1 × 10^6^ cells/mL) were cultured for 10–14 days in a hematopoietic progenitor medium (Table 2) for myeloid cell expansion. Partial medium change was performed every 3 days. In the final step, CD45^+^KIT^high^ immature MCs were FACS sorted as described below and further cultured in MCs’ maturation medium (Table 2) for 60–70 days (≤1 × 10^6^ cells/mL). Partial medium change was performed every 3–4 days. At the end of this step, MCs were referred to as mature MCs.

**Table 2 ijms-24-05275-t002:** Media composition for iPS cell differentiation toward mast cells.

					Spin EB (96-Well Format)			Hematopoietic Progenitors (10 cm Dish)	Mast Cell Maturation (6-Well Plate)
	Component	Final Concentration	Cat. N°	Brand	Day 0–2	Day 2–7	Day 8–14	Day 0–14	Day 0–60
BASAL MEDIUM	IMDM	50%	12440-053	Gibco					
	Ham’s F-12	50%	11765-054	Gibco					
	Chemically defined lipid concentrate	1%	11905-031	Gibco					
	GlutaMAX	2 mM	35050-038	Gibco					
	1-thioglycerol	400 µM	M1753	Sigma					
	BSA detox.	0.5%	A8022	Sigma					
	Albiomin 20%	0.5%	B235258	Biotest					
	L-ascorbic acid	50 µg/ml	72132	Stemcell Technologies					
	h-Transferrin	6 µg/ml	T0665	Sigma					
SUPPLEMENTS	Y-27632	10 µM	Ab120129	Abcam					
	BMP-4	10 ng/ml	130-111-168	Miltenyi					
	bFGF-2	10 ng/ml	100-18B	Peprotech					
	VEGF	10 ng/ml	100-20	Peprotech					
	SCF	50 ng/ml	130-095-745	Miltenyi					
	IL-3	30 ng/ml	200-03	Peprotech					
	Flt3L	50 ng/ml	300-19	Peprotech					
	hyper-IL-6	10 ng/ml		Fisher et al., 1997 [38]					

### 4.3. Magnetic Activated Cell Sorting (MACS)

The iPS cell-derived hematopoietic cells were harvested on day 14 of Spin EB differentiation and passed through a 40 µm cell strainer (Corning). After centrifugation at 350× *g* for 5 min, cells were resuspended in MACS/FACS buffer (5% FCS, 2 mM EDTA in PBS, all from Gibco, USA). Cells were subjected to MACS using the human CD34 MicroBead Kit and LS MACS Columns (both from Miltenyi Biotec, Bergisch Gladbach, Germany) following the manufacturer’s instructions.

### 4.4. Fluorescence Activated Cell Sorting (FACS)

FACS analysis was performed with a BD FACS Canto II (BD Bioscience, USA). The antibodies used for FACS are listed in Table 3. Briefly, 1–2 × 10^5^ cells pre-washed in MACS/FACS buffer were incubated with the diluted antibody solution (dilutions were performed in MACS/FACS buffer and the antibodies used, and respective dilutions are listed in Table 3) and incubated for 30 min at 4 °C, protected from light. Cells were washed once and resuspended in 500 µL MACS/FACS and subjected to FACS analysis. FACS sorting was performed with a FACS Aria II 3L (BD Bioscience, USA). Sorted cells were further cultured in an MCs’ maturation medium for 60–70 days. Data analysis was performed with FlowJo™ Software (BD Life Sciences, USA) and GraphPad Prism. 

### 4.5. Histochemistry and Immunohistochemistry

Cytospin preparations were performed with 1–2 × 10^5^ iPS cell-derived CD34^+^ HSPC or with 1–2 × 10^5^ iPS cell-derived MCs at distinct stages of differentiation using the Cytospin 4 Centrifuge (Thermo Scientific, USA). CD34^+^ HSPC staining was performed (Diff–Quick staining) after methanol fixation following the manufacturer’s instructions. MCs were stained with acidic toluidine blue after methanol fixation.

For tryptase staining, cytospin preparations with 1–2 × 10^5^ mature MCs were performed as described above and fixed with acetone. Immunohistochemical staining against MC tryptase was performed with Flex Kit DAKO/Agilent (DAKO, Carpinteria, CA, USA). Samples were incubated with an anti-human tryptase antibody (Table 3) for 30 min, followed by the application of a staining enhancer. Next, secondary antibody staining was performed, followed by visualization with horseradish peroxidase and DAB (DAKO). Counterstaining was performed with hematoxylin (DAKO).

### 4.6. IL-4 Stimulation of MC

Mature iPS cell-derived MCs (1–2 × 10^5^ cells/mL) were treated for 4 days with 10 ng/mL IL-4 (Peprotech, UK) in MCs’ maturation medium supplemented with SCF and hyper IL-6 (Table 2). After the incubation period, MCs were subjected to flow cytometry analysis as described above.

### 4.7. MC Degranulation Assay

MC degranulation assay was performed as described by Kuehn and colleagues [55]. Briefly, mature MCs were starved for 4 h in MCs’ maturation medium (Table 2) without SCF and hyper IL-6 supplementation, followed by stimulation with 100 ng/mL biotinylated IgE (DIA HE1B) for 16 h at 37 °C with 5% CO_2_. Cells were washed 3× with HEPES buffer, and 3 × 10^5^ cells were seeded in 90 µL of HEPES buffer per well in a 96-well plate. Cells were then incubated with 0.1 µg/mL DNP-HSA (Dinitrophenyl-human serum albumin conjugate, Sigma–Aldrich, USA) for 30 min at 37 °C. Secreted hexosaminidase activity was measured by incubating culture supernatant with PNAG (p-nitrophenyl N-acetyl-β-D-glucosamine, Sigma–Aldrich) solution (3.5 mg/mL) for 90 min at 37 °C. Enzymatic activity was measured by absorbance at 405 nm with a reference filter at 620 nm.

### 4.8. Global Gene Expression Analysis

Mature *KIT* D816V (n = 2) and control (n = 2) MCs were harvested by centrifugation (300× *g*, 5 min), and RNA was extracted via RNeasy Kit from Qiagen, Hilden, Germany following the manufacturer’s protocol. RNA libraries were prepared for sequencing using standard Illumina protocols with rRNA depletion (NEBNext rRNA Depletion Kit), NEBNext Ultra II Directional RNA Library Prep Kit, and NextSeq 500/550 Mid Output Kit v2.5, followed by sequencing on Illumina NextSeq 500. Quality control, alignment, and gene count extraction were performed with the NextFlow pipeline (v20.01.0) for RNAseq (v1.4.2). RNAseq count data were analyzed in R (v4.2.2). TMM normalization was applied with edgeR (v3.40.0), and the limma-trend pipeline (v3.54.0) was used to perform differential gene expression analysis. Finally, the R packages fgsea (v1.24.0), DOSE (v3.24.2), enrichplot (v1.18.1), ComplexHeatmap (v2.14.0), and gprofiler2 (v0.2.1) were used to perform and visualize gene-set enrichment and functional analysis.

### 4.9. RT-qPCR

RNA isolation of iPS cell-derived MCs was performed with NucleoSpin RNA Kit (Macherey–Nagel, Düren, Germany), and cDNA synthesis was performed with MultiScribe reverse transcriptase (High-Capacity cDNA Reverse Transcriptase Kit, Thermo Fisher Scientific, USA). For RT-qPCR, the FAST SYBR Green master mix (Thermo Fisher Scientific) was used, and runs were performed using a StepOnePlus Real-Time cycler. Primers (Eurofins Genomics, Ebersberg, Germany) are listed in Table 4. Data analysis and heatmap plots were conducted with GraphPad Prism and MeV-Multiple Experiment Viewer (http://mev.tm4.org/, accessed on 12 December 2022), respectively.

### 4.10. Transmission Electron Microscopy

The iPS cell-derived MCs were washed once with PBS and fixed with 3% glutaraldehyde for at least 2 h at RT. Next, cells were embedded in 5% low melting agarose (Merck, Darmstadt, Germany), followed by washing in 0.1 M Soerensen’s phosphate buffer (Merck) and post-fixed in 25 mM sucrose buffer (Merck) containing 1% OsO4 (Roth, Karlsruhe, Germany). Samples were dehydrated by performing an ascending ethanol series repeating the last step three times (30, 50, 70, 90, and 100% ethanol; 10 min each step). Dehydrated samples were incubated in propylene oxide (Serva, Heidelberg Germany) for 30 min, in a mixture of EPON resin (Serva) and propylene oxide (1:1) for 1 h, and in pure EPON for 1 h. EPON polymerization was conducted at 90 °C for 2 h. Ultrathin sections of 70–100 nm were performed with a Reichert Ultracut S ultramicrotome (Leica, Wetzlar, Germany) equipped with a diamond knife (Leica) and picked up on copper–rhodium grids (Plano, Wetzlar, Germany). Contrast enhancement was performed by staining with 0.5% uranyl acetate and 1% lead citrate (both EMS, Munich, Germany). Samples were viewed at an acceleration voltage of 60 kV using a Zeiss Leo 906 (Carl Zeiss, Jena, Germany) transmission electron microscope. Image processing and analysis were performed using ImageJ [56].

## Figures and Tables

**Figure 1 ijms-24-05275-f001:**
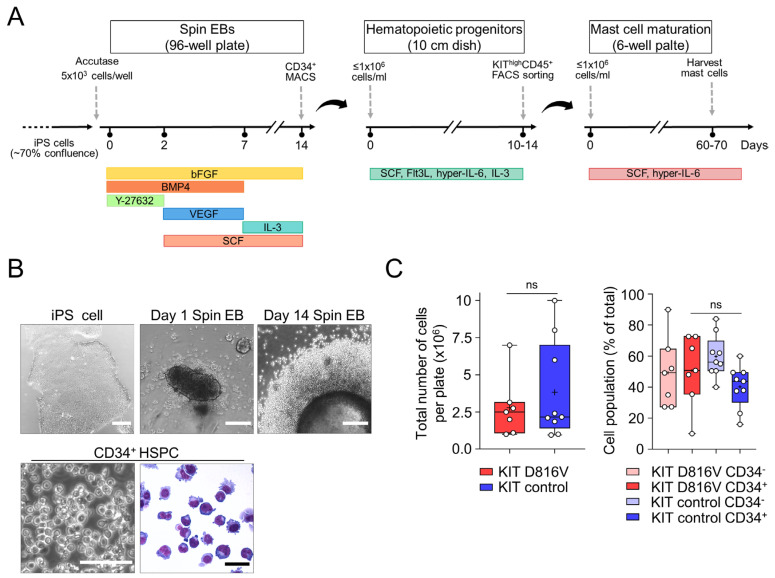
Differentiation of patient-derived iPS cells toward MCs. (**A**) Schematic overview of the protocol for differentiation of iPS cells toward the hematopoietic lineage and MCs. iPS cells are committed toward the mesoderm lineage through the formation of embryoid bodies (EB) using the spin-EB method and a defined cytokine cocktail. After 14 days, CD34^+^ cells are isolated by magnetic activated cell sorting (MACS), followed by myeloid lineage expansion with SCF, IL-3, hyper-IL-6, and Flt3L. Immature MCs are isolated by flow cytometry cell sorting and further cultivated in SCF and hyper-IL-6 for approximately 60 days. (**B**) Exemplary phase contrast microscopy images of feeder-free iPS cell culture (**upper left**, scale bar: 250 µm), day 1 spin-EB (**upper middle**, scale bar: 250 µm), and day 14 spin-EB showing hematopoietic cells budding-off from the EB (**upper right**, scale bar: 250 µm). After CD34^+^ MACS, a homogeneous population of hematopoietic stem/progenitor cells (HSPC) is obtained (**bottom panels**; **left**: phase contrast microscopy of CD34^+^ HSPC, scale bar: 250 µm, **right**: Diff–Quick stain of CD34^+^ HSPC, scale bar: 25 µm). (**C**) The total number of hematopoietic cells obtained per 96-well plate after 14 days of differentiation in the spin-EB step is shown (left graph). *KIT* D816V and control iPS cells generated a similar number of hematopoietic cells (Welch’s *t*-test, *p* = 0.4512, *KIT* D816V, n = 7; control, n = 9). The CD34^+^ population on day 14 is shown on the right. *KIT* D816V and control iPS cells generated a similar number of CD34^+^ HSPC (Welch’s *t*-test, *p* = 0.27, *KIT* D816V, n = 7; control, n = 9). For the graphs in (**C**), box and whiskers plots show minimum and maximum values, and “+” indicates the mean value. ns indicates not significant.

**Figure 3 ijms-24-05275-f003:**
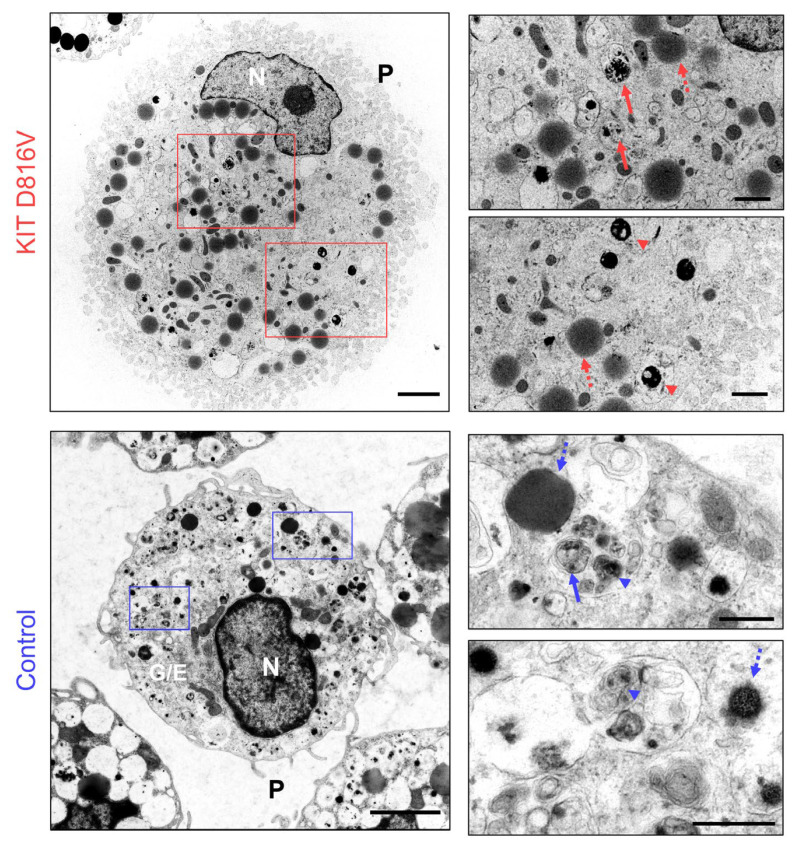
Ultrastructural characterization of iPS cell-derived MCs. Transmission electron microscopy (TEM) of *KIT* D816V MCs (**top panels**, scale bar left panel: 2500 nm, **right panels**: 1000 nm) and control MCs (**bottom panels**, scale bar left panel: 2500 nm, **right panels**: 500 nm). Cytoplasmic thin projections (P) are observed, as well as MC characteristic membrane-delimited secretory granules with dense cores in a translucent surrounding (arrows), larger electron dense/light granules (dashed arrows), and granules displaying electron-dense “scroll-like” arrangement (arrow heads). Nucleus (N) and Golgi/endoplasmic reticulum (G/E) are indicated.

**Figure 4 ijms-24-05275-f004:**
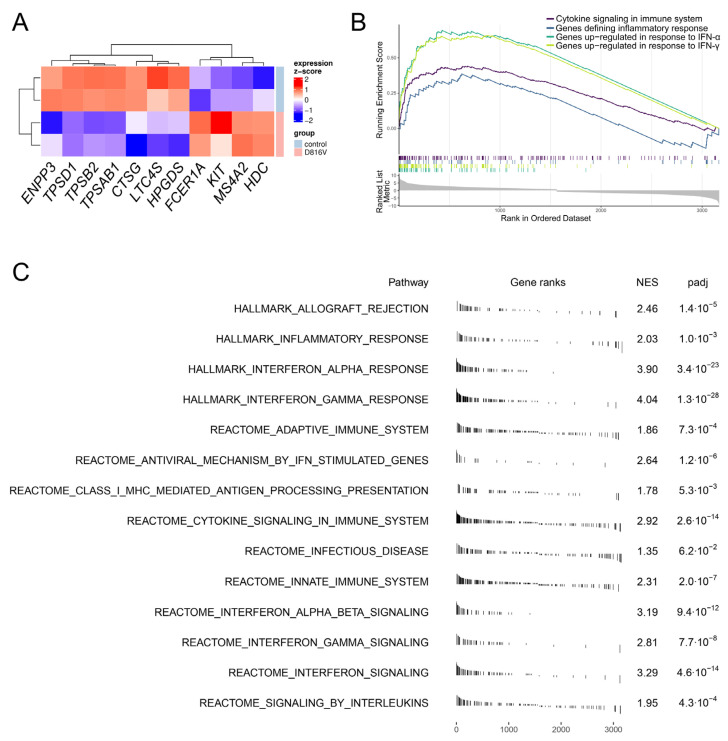
*KIT* D816V iPS cell-derived MCs display inflammatory and immune response transcriptional profile. (**A**) Global gene expression analysis of *KIT* D816V (n = 2) and control (n = 2) iPS cell-derived MCs confirmed MC identity as 11 out of 14 MC signature genes were highly expressed (please also see Appendix A). *KIT* D816V MCs showed higher expression of *KIT* and FcER1 subunits *FCER1A* and *MS4A2*. Gene expression z-score is plotted as a heatmap. (**B**) Gene set enrichment analysis (GSEA) of iPS cell-derived MC transcriptome revealed an upregulation of pathways related to cytokine signaling in the immune system, inflammatory, and IFN-α/β/γ responses in *KIT* D816V cells. (**C**) Enriched pathways in *KIT* D816V MC expression profile identified by GSEA. *KIT* D816V MCs show upregulated pathways related to innate and adaptive immune system as well as inflammatory response, IFN, and cytokine signaling, among others. The gene set names, with respective gene set collection names, are listed with the normalized enrichment score (NES) and adjusted *p*-value (padj).

**Figure 5 ijms-24-05275-f005:**
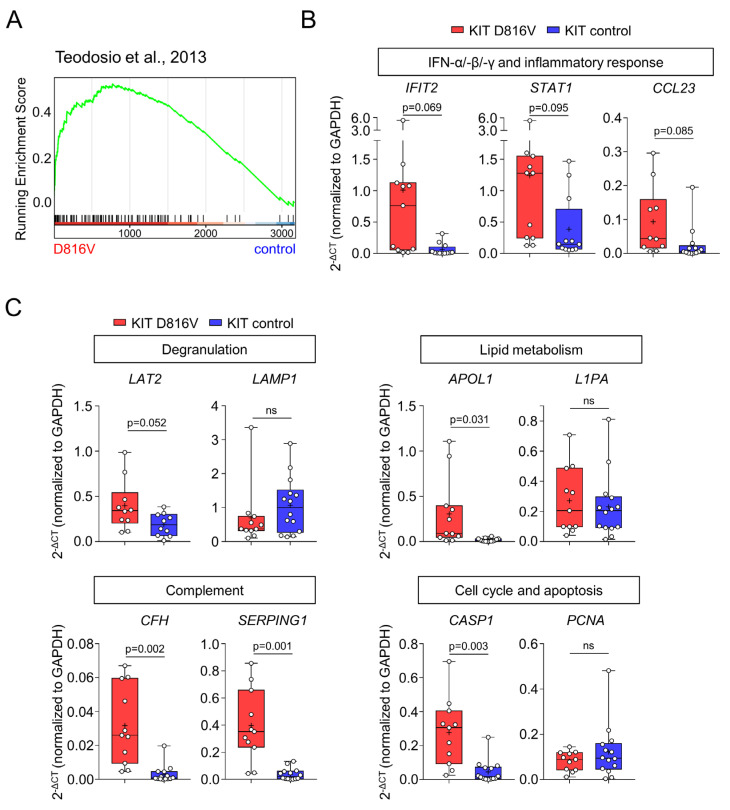
*KIT* D816V iPS cell-derived MCs recapitulate the transcriptional profile observed in primary SM MCs. (**A**) Gene sets, identified to be deregulated in primary BM MCs of SM patients in the work of Teodosio and colleagues [48], are enriched in *KIT* D816V iPS cell-derived MCs, as shown by GSEA. (**B**) Validation of the GSEA by RT-qPCR. Upregulation in IFN signaling pathways was validated by *IFIT2* and *STAT1* gene expression, and upregulated inflammatory response pathway was validated by *CCL23* gene expression. (**C**) RT-qPCR validation of additional upregulated pathways identified in our GSEA and by Teodosio and colleagues. For (**B**,**C**), box and whiskers plots show minimum and maximum values; “+” indicates the mean value, and statistical analysis is performed with Welch’s *t*-test (*KIT* D816V n = 10–11, control n = 10–14). ns indicates not significant.

**Table 3 ijms-24-05275-t003:** List of antibodies.

Antibody	Specificity	Company	Cat. N°	Dilution	Assay
CD2-FITC	human	BD Biosciences	556608	1:50	FACS
CD25-PE	human	BD Biosciences	567214	1:100	FACS
CD30-PE	human	BD Biosciences	550041	1:100	FACS
CD33-APC	human	Miltenyi Biotec	130-113-345	1:100	FACS
CD45-APCVio770	human	Miltenyi Biotec	130-113-677	1:200	FACS
CD117 (KIT)-PE-Cy7	human	eBioscience	25-1178-42	1:100	FACS
FcER1a-PE	human	BD Biosciences	566608	1:100	FACS
Tryptase	human	Dako	M7052	1:200	IH

**Table 4 ijms-24-05275-t004:** List of RT-PCR primers.

Oligo Name	GC-Content	Tm	Sequence (5′->3′)	Length [mer]
*APOL1*_FRW	55	62.1	GCTTTGCTGAGAGTCTCTGTCC	22
*APOL1*_REV	52	62.4	GGGCTTACTTTGAGGATCTCCAG	23
*CASP1*_FRW	55	59.4	GCTTTCTGCTCTTCCACACC	20
*CASP1*_REV	50	57.3	TCCTCCACATCACAGGAACA	20
*CCL23*_FRW	50	57.3	ATGCTTGTTACTGCCCTTGG	20
*CCL23*_REV	55	59.4	GGGTCATCTGAGGACCAATC	20
*CFH*_FRW	45	58.4	CTGATCGCAAGAAAGACCAGTA	22
CFH_REV	45	58.4	TGGTAGCACTGAACGGAATTAG	22
*GAPDH*_FRW	56	56	GAAGGTGAAGGTCGGAGT	18
*GAPDH*_REV	45	58	GAAGATGGTGATGGGATTTC	20
*IFIT2*_FRW	52.6	65	AGCGAAGGTGGCTTTGAGA	19
*IFIT2*_REV	55	65	GAGGGTCAATGGCGTTCTGA	20
*LAMP1*_FRW	50	57.3	TGAAAAATGGCAACGGGACC	20
*LAMP1*_REV	55	59.4	ATGAGCTGGACGCTGTAACG	20
*LAT2*_FRW	55	59.4	CAGCATCCATCCATCAGTGG	20
*LAT2*_REV	58	58.8	TTCTTGGTCTGCCCTAGGC	19
*LIPA*_FRW	55	59.4	CTAGAATCTGCCAGCAAGCC	20
*LIPA*_REV	45	55.2	TGTGCCTTAACCGAATTCCT	20
*MRGPRX2*_FRW	52	59.8	AGTCCCAGGAAAGCACTTCTC	21
*MRGPRX2*_REV	55	59.4	AGGGTCTCCTTGCCACAAAG	20
*PCNA*_FRW	42	59.3	GAACTGGTTCATTCATCTCTATGG	24
*PCNA*_REV	35	58.5	TGTCACAGACAAGTAATGTCGATAAA	26
*SERPING1*_FRW	55	59.4	TGAAGCTCTACCACGCCTTC	20
*SERPING1*_REV	55	59.4	TGGTGGACACAGGTGAAGTC	20
*STAT1*_FRW	55	59.4	TGTATGCCATCCTCGAGAGC	20
*STAT1*_REV	55	59.4	AGACATCCTGCCACCTTGTG	20

## Data Availability

Gene expression profile data can be accessed through the accession number GSE223883, token cnwhgcugtboxrer. Further information or resources may be provided upon direct contact with Martin Zenke (martin.zenke@rwth-aachen.de).

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
