# Peer review of "KIT D816V Mast Cells Derived from Induced Pluripotent Stem Cells Recapitulate Systemic Mastocytosis Transcriptional Profile"

_ijms, 2023, doi:10.3390/ijms24065275_

Round 1
Reviewer 1 Report
In their manuscript titled: “KIT D816V mast cells derived from induced pluripotent stem cells recapitulate systemic mastocytosis transcriptional profile”, de Toledo et al have the potential of induced pluripotent stem (iPS) cells to give rise to mast cells (MC). Of interest, the authors have also used this method to derive MC from systemic mastocytosis (SM) patient-specific iPS cell lines carrying the KIT D816V mutation. They have then analyzed the KIT D816V+ MC generated through this technique and compared it to KIT D816V-negative MC obtained by the same method. Of note, it appears from the data presented here that IPSC-derived KIT D816V+ MC seem to recapitulate SM disease features, with abnormal maturation kinetics, activated phenotype, expression of CD25 and CD30, and upregulation of the expression of KIT (CD117) and of a functional high affinity receptor for IgE (FceRI). In addition, the KIT D816V+ MC appear to have a transcriptional signature different of the one of their KIT wild-type (WT) counterpart generated by the same technique. Indeed, it appears that the KIT D816V+ MC are characterized by upregulated expression of several innate and inflammatory response genes.
These data are obviously of potential interest for all researchers working on the physiopathology of systemic mastocytosis as well as of other mast cell disorders. Indeed, they tend to show that human iPS cell-derived MC could be a reliable, inexhaustible, and close-to-human tool for disease modeling and pharmacological screening to explore novel MC therapeutics.
Anyway, I have 2 major points of discussion to point regarding this manuscript:
1) Usually, the KIT D816V mutation is considered to be the main driver of the pathology in indolent systemic mastocytosis (ISM), which contrasts to more advanced variants of the disease (AdvSM) where, as underlined by the authors themselves, additional genetic defects are found which aggravate the prognosis of the disease. Among the most frequently encountered additional defects are TET2 mutants, but also SRSF2, ASXL1, RUNX1 and RAS defects. Looking at Table 1, for patient 1, it appears that the patient had probably NRAS and TET2 mutations in addition to KIT D816V mutant. However, for this patients the authors have used IPSC-derived KIT WT MC with these 2 additional mutations, and IPSC-derived KIT D816V+ MC WITHOUT these 2 mutants, but with a mutant never found in AdvSM, i.e. NFE2. This means 1) that their IPSC-derived KIT D816V+ MC are not representative of what is usually found in AdvSM, while their IPSC-derived KIT WT MC are not representative of “normal” MC. It is thus difficult to fully trust in the data generated claiming that the KIT D816V+ MC are characterized by upregulated expression of several innate and inflammatory response genes as compared to their KIT WT counterpart.
2) Using a cord-blood derived human KIT WT, SCF-dependent MC line (ROSA KIT WT), Saleh et al (Blood, 2014 Jul 3;124(1):111-20) have demonstrated that the simple introduction of the KIT D816V mutant gene in such cell line is able to render it SCF-independent. Here, in the present report, nothing is known about the SCF-dependence status of the IPSC-derived KIT D816V+ MC generated as compared to IPSC-derived KIT D816V-negative MC. This reviewer should be interested in knowing what will happen if the IPSC-derived KIT D816V+ MC are shifted in a culture medium deprived of SCF. As well, Saleh et al, in the same manuscript, reported that the introduction of the KIT D816V mutant gene in the ROSA KIT WT cell line induced a tumorigenic phenotype of the cell line in vivo in NSG mice. This reviewer should be interested to know whether IPSC-derived KIT D816V+ MC described by the authors are tumorigenic in vivo (in NSG mice, in NSGhSCF mice?). Indeed, since the authors claim here that that IPSC-derived KIT D816V+ MC seem to recapitulate SM disease features, it would be nice to know if they can recapitulate such features up to engraftment capability in vivo.
Minor points:
1) In the manuscript, some figures are way too small to be easily readable: Figure 1, Figure 2, Figure 4.
2) Please adhere to the international nomenclature: KIT D816V. If speaking of the gene, KIT has to be in italics. If referring to the protein, KIT is in straight right letters.
Reviewer 2 Report
De Toledo et al describe KIT D816V mutated mast cells from iPS cells with transcriptional profile of SM disease. The manuscript is well written and contains key references in the field however some issues remain to be clarified, e.g. demonstration of KIT D816V mutation, tryptase stain of the mast cells, and tryptase secretion in the iPS mast cells.
Major comments
Please include data to show that KIT D816V mutation is indeed present in the end product iPS derived mast cells and the variant allele frequency, is this 50% thus one allele in each cell.
Please also include data on the other myeloid mutations that were found in the patients from which the iPS lines were produced, are they present in the iPS MC and at what allele frequency?
Please include morphology of iPS mast cells with tryptase stain ensuring tryptase in the granulae to complement Figure 3 and suppl fig 2.
Figure 4a. The iPS lines express more KIT however do not express any of the tryptase genes. Please comment and please provide morphology with tryptase stain confirming that the iPS cells do produce tryptase in their granulae, and secrete tryptase upon stimulation e.g. IgE crosslinking like mature mast cells.
Line 108-115 and figure 1a, Table 2. Please describe the starting product for your embryid body EB formation thus the steps preceding step 1, how you establish your iPS line from the patient cells, what starting cell you use from the patients and how you come to step 1.
Minor suppl fig 1. Y axis measurement “beta hexosaminidase activity in % of total activity” . Please explain what this is measuring, and explain “% of total activity”. Did you measure tryptase in the media after IgE stimulation?
Minor suppl fig 3. Global gene expression in iPS derived mast cells. On thy Y axis is “TPM” and “TPM x103. Please explain the TPM or change to log 2 fold change compared to control or other established entity.
Please comment on the iPS cell lines from patient 1 ASM, how the iPS KIT D816V mutation containing lines do not also contain NRAS that seems to be a somatic stem cell mutation of the ASM patient. In addition, patient 2 and patient 3 were MCL patients, were they not subjected to target sequencing panels, surely they must have had other somatic mutations as well? Please comment on this.
Round 2
Reviewer 1 Report
No comments
Reviewer 2 Report
The manuscript is now acceptable for publication.
Note there is a typo in Fig 1a right side of figure "palte" should be "plate".